# DUODUO CLIP: EFFICIENT 3D UNDERSTANDING WITH MULTI-VIEW IMAGES

**Han-Hung Lee[1,*]  Yiming Zhang[1,*]  Angel X. Chang[1,2]**
[1]Simon Fraser University,  [2]Canada CIFAR AI Chair, Amii
{hla300, yza440, angelx}@sfu.ca
https://github.com/3dlg-hcvc/DuoduoCLIP

## ABSTRACT

We introduce Duoduo CLIP, a model for 3D representation learning that learns shape encodings from multi-view images instead of point clouds. The choice of multi-view images allows us to leverage 2D priors from off-the-shelf CLIP models to facilitate fine-tuning with 3D data. Our approach not only shows better generalization compared to existing point cloud methods, but also reduces GPU requirements and training time. In addition, the model is modified with cross-view attention to leverage information across multiple frames of the object which further boosts performance. Notably, our model is permutation invariant to the order of multi-view images while being pose-free. Compared to the current SOTA point cloud method that requires 480 A100 hours to train 1 billion model parameters we only require 57 A5000 hours and 87 million parameters. Multi-view images also provide more flexibility including being able to encode objects with a variable number of images, and performance scales when more views are used. In contrast, point cloud based methods require an entire scan or model of the object. We showcase this flexibility with benchmarks from images of real-world objects. Our model also achieves better performance in more fine-grained text to shape retrieval, demonstrating better text-and-shape alignment than point cloud based models.

## 1 INTRODUCTION

Learning representations for 3D shapes is useful for many applications in object generation, shape understanding, grounding and robotics. Recently, there is also growing interest in aligning 3D representations to text, to enable text-to-3D shape retrieval (Ruan et al., 2024; Tang et al., 2023; Wang et al., 2023) and generation (Chen et al., 2019), or as a basis for 3D feature maps that can be queried with text (Huang et al., 2023a; Jatavallabhula et al., 2023; Liu et al., 2023a).

Early work in this domain attempted to train such aligned models on limited data (Chen et al., 2019; Ruan et al., 2024; Tang et al., 2023). More recent work (Liu et al., 2023a; Xue et al., 2023) leveraged large pre-trained vision-language models (e.g. CLIP (Radford et al., 2021)). Larger 3D datasets like Objaverse (Deitke et al., 2023) enabled training of methods that can handle more diverse object categories (Liu et al., 2023a; Qi et al., 2024; Xue et al., 2024; Zhou et al., 2024). However, more and more GPU

Table 1: Comparison of training time and GPU used for our method and recent point cloud based methods.

| Method | GPU | Time |
|---|---|---|
| OpenShape Liu et al. (2023a) | 1×A100 (80GB) | 300 hr |
| Uni3D Zhou et al. (2024) | 24×A100 (40GB) | 20 hr |
| RECON++ Qi et al. (2024) | 8×A800 (80GB) | 1 day |
| Ours (Full) | 4×A40 (48GB) | 14.3 hr |
| Ours (6 layers) | 4×A5000 (24GB) | 14.3 hr |

resources are required for training such methods. These works (Liu et al., 2023a; Qi et al., 2024; Xue et al., 2024) predominantly rely on point clouds as their 3D representation. Point clouds are limited in resolution and have a domain gap with real-world images, and thus these point cloud-based methods can not effectively leverage 2D priors from CLIP. Uni3D (Zhou et al., 2024) proposed a training regime to incorporate 2D priors with point clouds as input. However, their model requires significant scaling (1B parameters) for the best performance. In this work, we show that our Duoduo CLIP approach which uses multi-view images as the shape representation allows more easily adapting off-the-shelf 2D representation models while enabling more efficient training. As shown in Fig. 1b, our model not only has much stronger generalization on shapes that were not seen during training, but also significantly lowers the compute resources required for training (see Tab. 1).

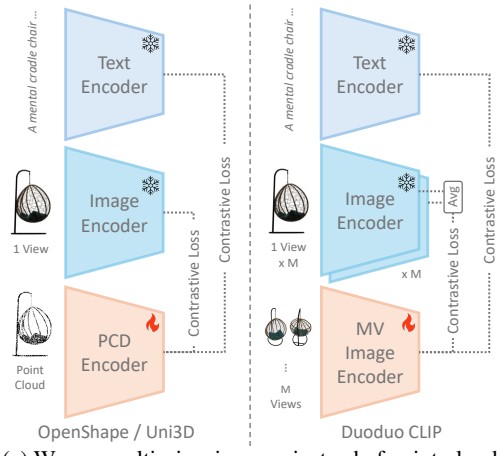

(a) We use multi-view images, instead of point-clouds, to represent the shape.

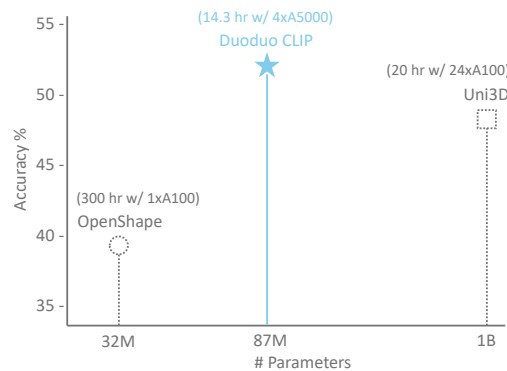

(b) Accuracy on Objaverse-LVIS with all models trained on *Ensembled (no LVIS)*. Our Duoduo CLIP is fast to train, relatively small while also achieving the best performance.

Figure 1: Duoduo CLIP is a model for 3D shape understanding that represents the shape with a multi-view images encoder instead of a point-cloud (PC) encoder. This makes our model more accurate and efficient than PC methods (OpenShape Liu et al. (2023a), Uni3D Zhou et al. (2024)).

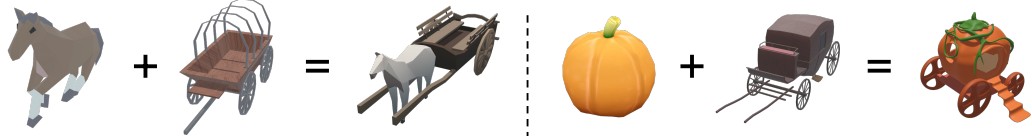

Figure 2: We blend concepts from two distinct objects by identifying the Objaverse shape with the embedding that maximizes similarity to both objects simultaneously. This shows that our model has learned a latent space capable of combining geometric features from the mixed objects or capturing their semantic attributes. For more details see App. A.4.2.

Multi-view images also offer greater flexibility. For example, in real-world datasets like ScanNet (Dai et al., 2017) or ScanObjectNN (Uy et al., 2019) a 3D mesh is first reconstructed from densely captured RGBD data from which 3D point clouds are sampled. This process introduces errors and artifacts, in contrast to sparse multi-view images which can be directly sub-sampled from the RGB video stream. Our model takes images directly as input, making it plug-and-play for any method using image features for encoding 3D objects, which is particularly important for real-time robotics applications with sparse views (Gadre et al., 2023; Garg et al., 2024; Shridhar et al., 2022; Wang et al., 2024). Additionally, previous methods (Liu et al., 2023a) that only train on synthetic objects have limited performance on real-world objects. Our method can be trained on both real photos and reconstructed objects and outperforms prior work on real-world objects. Performance can further be boosted with additional multi-view images when point clouds are not available, demonstrating our model's potential for scaling as more data becomes available.

In summary, our Duoduo CLIP model provides the following: **Simpler model.** We average embeddings across views of an object to obtain 3D representations, while modifying the attention layers to span multiple frames of the shape to enable information flow across frames in the model. **Better efficiency.** We explore several strategies for efficiently fine-tuning off-the-shelf CLIP models, including training only the attention layers and selectively choosing the number of layers to train without hurting performance. **More flexible representation.** We show the benefits of multi-view images as a more flexible 3D representation over point clouds on real-world objects. Furthermore, we demonstrate better text to shape retrieval with more fine-grained descriptions, and that by maximizing similarity to two different shape embeddings, our learned embedding space can also be used to blend concepts (see Fig. 2).

## 2 RELATED WORK

There is much work on 3D representation learning using supervised training (Qi et al., 2017; Qian et al., 2022) or using self-supervised learning (with reconstruction loss (Chen & Zhang, 2019),

contrastive learning (Xie et al., 2020), masked prediction (Yu et al., 2022b), and others). Here we focus on work that learns aligned representation spaces between text and either 2D or 3D.

**2D-text joint representation.** Modeling multiple modalities through a shared representation has been explored through early works by aligning images and text embeddings (Faghri et al., 2018; Frome et al., 2013; Kiros et al., 2014; Weston et al., 2011). With the introduction of the InfoNCE loss (Oord et al., 2018), ConVIRT (Zhang et al., 2022b) proposed using contrastive learning with InfoNCE style loss to learn joint text and image embeddings. This was popularized by follow-up work such as CLIP (Radford et al., 2021) and ALIGN (Jia et al., 2021), which demonstrated that by scaling up training on a large-scale dataset, the learned embeddings can be useful for many downstream tasks such as classification, retrieval, grounding and generation. Followup work (Jia et al., 2021; Mu et al., 2022; Yang et al., 2022; Yu et al., 2022a; Zhai et al., 2022) has since improved upon the formulation or training. There have also been attempts such as MERU (Desai et al., 2023) that use a hyperbolic embedding space to better model the hierarchical relations between text and images.

**3D-text joint representation.** Text2Shape (Chen et al., 2019) was one of the first works that introduced a captioned text and 3D dataset and used metric learning to learn an aligned space between text and 3D shapes. Follow-up work investigated improvements to the aligned representation for text-to-shape retrieval with triplet loss (Tang et al., 2023) and contrastive losses (Ruan et al., 2024; Wang et al., 2023; Xue et al., 2023). Some prior work used of CLIP to align the embeddings (Ruan et al., 2024; Thomason et al., 2022; Xue et al., 2023). ULIP (Xue et al., 2023) used frozen CLIP text-image encoders, and trained only the point-cloud encoder. However, generalization ability was limited by the available dataset size at the time. With the introduction of larger scale 3D datasets such as Objaverse (Deitke et al., 2023), and the use of LVLMs for captioning 3D datasets (Liu et al., 2023a; Luo et al., 2024), more recent work (Liu et al., 2023a; Xue et al., 2024) trained with larger datasets of paired text and 3D shapes. OpenShape (Liu et al., 2023a) and ULIP-2 (Xue et al., 2024) were among the first to use such datasets to train a point cloud encoder aligned with text using contrastive learning. TAMM (Zhang et al., 2024) leverages additional adapters to better align with the pre-trained CLIP models. Uni3D (Zhou et al., 2024) exploits 2D priors within pre-trained large-scale vision models and maps point cloud patches to image patches within pre-trained ViTs to enable the scaling of even larger models. VIT-LENS (Lei et al., 2024) follows a similar pipeline by training a point cloud embedding layer and perceiver model to map to a frozen CLIP model. RECON++ (Qi et al., 2024) similar to RECON (Qi et al., 2023) employs both contrastive and MAE (He et al., 2022) reconstruction losses with an additional multi-view image matching cost. These recent works typically rely on frozen text and image encoders, and all used point-cloud encoders. Our work shows that taking embeddings of multi-view images is more efficient and works better with real-world images. Note that Gao et al. (2024) combines point clouds with optional multi-view embeddings from a frozen CLIP encoder, which lacks inter-view context, and does not support arbitrary views or poses, resulting in lower performance at the same number of views while also requiring point cloud.

**Multi-view images for text-3D representations.** Multi-view images have long been used to represent 3D shapes (Bradski & Grossberg, 1994; Hamdi et al., 2021; Shen et al., 2003; Su et al., 2015; Wei et al., 2020), with recent work (Lin et al., 2024; Song et al., 2023) using multi-view images for shape representation by aggregating CLIP features. While these works also use CLIP with multi-view images, our focus is whether using a trainable multi-view shape encoder as a drop-in-replacement for the point-cloud encoder in contrastive settings such as OpenShape will result in better shape representations for downstream tasks. These works do not focus on improved representation learning, but on how to use prompting to select better views or view aggregation and improve shape classification. There is also a line of work (Huang et al., 2023b; Zhang et al., 2022a; Zhu et al., 2023) that uses multi-view depth maps from point cloud to adapt CLIP models for 3D. However, the focus of these methods is still on point clouds and the models show limited performance on large-scale datasets (Liu et al., 2023a; Xue et al., 2024).

## 3  METHOD

Our Duoduo CLIP learns a shape representation by using contrastive learning to encode multi-view images into a pre-aligned text-image space. Our contrastive learning framework is similar to previous works (Liu et al., 2023a; Xue et al., 2023; 2024; Zhang et al., 2024; Zhou et al., 2024), with a shape encoder $E^S$ and frozen image $E^I$ and text $E^T$ encoders from a pre-trained CLIP model. The key

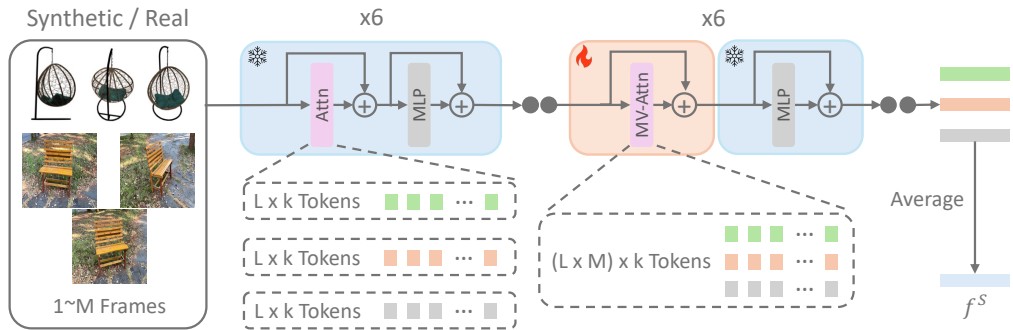

Figure 3: Our multi-view encoder takes a variable number $[1, M]$ of images as input and outputs a single-shape embedding. The first few layers of the ViT model are frozen (blue) and attention operates on the individual frames in parallel. For the latter layers, the attention layers are trainable (orange) and modified to attend over all $M$ views. The embeddings for each frame are averaged to get the final embedding. Note that only major components of the model are depicted.

difference is that we replace the point cloud shape encoder with one that uses multi-view images. Fig. 1a illustrates this difference, while Fig. 3 shows the details of our multi-view shape encoder.

To perform training, the dataset is organized into triplets of $(S_i, I_i, T_i)$, where $S$ is the shape representation, $I$ are rendered images of the 3D object and $T$ is the corresponding text description. Embeddings are calculated for each modality $f_i^S = E^S(S_i)$, $f_i^I = E^I(I_i)$ and $f_i^T = E^T(T_i)$. The asymmetric contrastive loss (Zhang et al., 2022b) between any two modalities is:

$$l_i^{a \to b} = -\log \frac{\exp(\langle f_i^a, f_i^b \rangle)/\tau}{\Sigma_{k=1}^N \exp(\langle f_i^a, f_k^b \rangle)/\tau} \tag{1}$$

where $a, b$ is any two modalities, $\tau$ is the temperature and $N$ is the batch size. Then, the full loss is:

$$L_{CON} = \frac{1}{4N} \Sigma_{i=1}^N (l_i^{S \to T} + l_i^{T \to S} + l_i^{S \to I} + l_i^{I \to S}) \tag{2}$$

Two important properties of our shape encoder for multi-view images include being able to handle arbitrary number of views as well as being permutation invariant to ensure the flexibility for the input images. Encoding of any number of frames with the model into a single embedding as well as a permutation invariant operation between multi-views are discussed in more detail below. Note that we also do not make assumptions about the poses of the input image, making our method pose-free.

**Multi-view shape encoder.** For our multi-view shape encoder, we use rendered images of the object. Note that the input shape and image representations are the same, i.e. $S_i = I_i = \{r_1, ..., r_m\}$, where $r_j$ is a rendered frame, and $m$ is the number of multi-views. $E^I$ and $E^S$ share the same network architecture, with the main difference between $E^I$ and $E^S$ is that $E^I$ is frozen, while we train $E^S$. Weights for $E^S$ are initialized from $E^I$. Empirically, we found it was useful to keep the contrastive losses $l_i^{S \to I} + l_i^{I \to S}$. We suspect that by keeping $E^I$ frozen and the contrastive loss with the original image embeddings, our $E^S$ can learn a good representation that is close to the text embeddings, but does not drift too much away from the initial image embeddings.

For both $E^S$ and $E^I$ we average the extracted embeddings for all $m$ views to produce a single latent representation for an object: $f_i^S = E^S(S_i) = \frac{1}{m} \Sigma_{j=1}^m \text{ViT}(S_i)[j]$, where ViT (Dosovitskiy et al., 2021) is the backbone encoder. Here we abuse the notation a bit, since our method allows interaction between views in the model, the index operation gets the corresponding embedding from the $j$-th frame's [CLS] token. We initialize $E_S$ initially from $E^I$. During training, we randomly sample 1 to $M$ views for each step so that the model is robust to a variable number of views.

**Multi-view attention.** We incorporate information flow between views by modifying the self-attention layers to attend over tokens across multiple views similar to recent multi-view aware methods like MVDream (Shi et al., 2023). The original self-attention (Vaswani et al., 2017) is given by $\text{Attention}(Q, K, V) = \text{softmax}(\frac{QK^T}{\sqrt{d_k}})V$, where $Q, K, V \in R^{L \times k}$, $L$ is the token length for a single image after patchify operations, and $k$ is the channel dimension. For cross-view attention we aggregate token features across $m$ multi-views (MV) so that $Q^{\text{MV}}, K^{\text{MV}}, V^{\text{MV}} \in R^{(L \times m) \times k}$ and the $\text{Attention}(Q^{\text{MV}}, K^{\text{MV}}, V^{\text{MV}})$ can be calculated without changing the parameters of the original

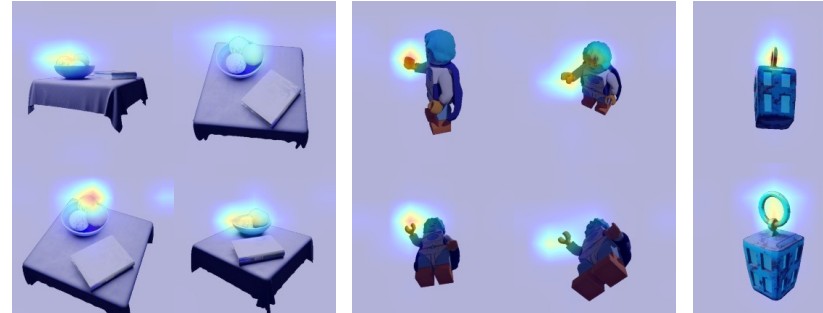

Figure 4: Attention maps are extracted from the first trainable MVA layer, capturing token-wise attention across all M views. A specific token (e.g., the plate) is queried to retrieve the corresponding row from the attention map, and its magnitudes are mapped onto the original images for visualization. The model has learned geometric correspondences of 3D shapes across different multi-views.

model. After attention, the views are separated and processed in parallel in other parts of the model. This mechanism helps the model reason between different views of the object and aggregate the most important information across all tokens. In Fig. 4, we visualize the attention across views corresponding to a selected token (see App. A.5.1 for details). From the figure, we see the multi-view attention layers learn to capture correspondence between object parts from different views.

**Trainable layers.** To reduce the memory needed for fine-tuning, we only train a portion of the self-attention layers while keeping the MLP layers after attention fixed. Since the non-trainable self-attention layers have only been trained to see patches of a single image, we only enable the multi-view attention layers for the unfrozen attention layers (see Fig. 3). Additionally, we find the choice of freezing the MLPs, helps in preventing overfitting and makes the model generalize better on unseen datasets (see App. A.2.2).

## 4 EXPERIMENTS

To compare our Duoduo CLIP to prior work, we conduct experiments showing shape classification on a collection of 3D assets (on Objaverse-LVIS in Sec. 4.2.1), real-world multi-view images (on MVImgNet in Sec. 4.2.2), and more fine-grained text-to-shape retrieval (Sec. 4.3). We also show image-to-shape retrieval results for out-of-distribution images (App. A.4.1) and concept mixing (App. A.4.2, where we retrieve shapes that maximizes similarity to two input images. We first describe the experimental setup including implementation details and the datasets we use.

### 4.1 EXPERIMENTAL SETUP

**Implementation details.** All of our models are trained with 16-bit mixed precision and a batch size of 1600 for 80 epochs. We use a learning rate of $5e-5$ with cosine annealing. At each training step we randomly sample 1 to 6 multi-views for a batch of objects. Ablations regarding the number of multi-views sampled during training can be found in App. A.2.1. The pretrained CLIP model used for initialization as well as the contrastive target is the ViT-B/32 CLIP model with checkpoint *laion2b_s34b_b79k* from the open source implementation OpenCLIP (Ilharco et al., 2021). See App. A.2.3 for a discussion of how initialization affects the performance. Additionally, we re-implemented OpenShape with 16-bit mixed precision training without projection layers for faster training in some baselines, indicated by OpenShape†.

**3D Datasets.** We follow OpenShape (Liu et al., 2023a) and train our model on a combination of four datasets consisting of Objaverse (Deitke et al., 2023), ABO (Collins et al., 2022), ShapeNet (Chang et al., 2015) and 3D-FUTURE (Fu et al., 2021). Text for the shapes consists of captions produced by image-captioning models and filtered text descriptions from metadata of the 3D objects crawled from the web (see Liu et al. (2023a) for details). In total, the combined dataset contains ∼874k shapes, with 46k shapes within the LVIS subset of Objaverse which are used for evaluation. As in OpenShape (Liu et al., 2023a), we consider two variations of the data: 1) *Ensembled* which contains all shapes, and 2) *Ensembled (no LVIS)* which excludes shapes from the LVIS subset.

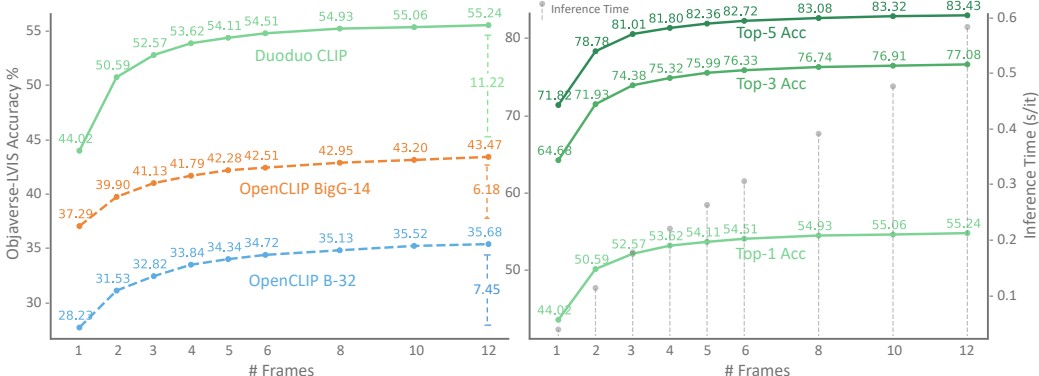

Figure 5: **Left.** We compare the Top 1 accuracies on Objaverse-LVIS for our model (Duoduo CLIP) and two pretrained CLIP models (zero-shot) for different number of frames. B-32 is the model we initialize from and train with and BigG-14 is the model used by OpenShape. The dashed lines show the difference between using 1 and 12 frames. **Right.** We show the Top1, Top3 and Top5 accuracies for our model for 1 to 12 frames. Gray dashed lines show the inference time(s) for 1 iteration and batch size 200 at different number of frames.

**Renderings.** We use images from Zero123 (Liu et al., 2023b), which provides renderings of most shapes from Objaverse from 12 random views using spherical sampling. The shapes not included in Zero123 (a few remaining objects in Objaverse and the other 3 datasets) are rendered using their Blender script to match the shapes in OpenShape. We render 12 views for all these objects. Additional details and effects of view selection can be found in App. A.1.2.

**Real-world data.** *Multi-view images.* We leverage MVImgNet (Yu et al., 2023), a dataset comprising multi-view images for 220k real-world objects across 238 categories. To focus on objects with sufficient views, we filter those with at least 12 views, evenly sampled, resulting in ~190k objects. For each view, we generate captions using BLIP-2 (Li et al., 2023). To facilitate comparisons with point cloud-based methods, we use the MVPNet subset, which includes ~87k objects with densely reconstructed point clouds. After preprocessing and filtering, we retain ~66k objects for the training split and ~16k for the validation set, spanning 180 categories. Importantly, the training and validation objects are disjoint but originate from the broader MVImgNet dataset. *Reconstructed 3D data.* To further evaluate our approach, we conduct experiments using objects from real-world scans. For instance, ScanObjectNN (Uy et al., 2019) provides point clouds derived from objects in SceneNN (Hua et al., 2016) and ScanNet (Dai et al., 2017). We evaluate on the test set, which contains 583 shapes across 15 categories, rendering 12 multi-view images directly from the point clouds (see App. A.3.1 for details). Additionally, we use the ScanNet validation set, consisting of 3825 objects across 17 classes, to perform classification tasks. For this, we crop multi-view images from the raw captured RGB frames in the dataset (details in App. A.3.2).

## 4.2 CLASSIFICATION EXPERIMENTS

### 4.2.1 OBJAVERSE-LVIS

We conduct classification experiments on Objaverse-LVIS to investigate the performance of different variants of Duoduo CLIP and compare our model to prior work. As default settings for our experiments in this section, we use 12 multi-views to encode an object for evaluation. And only the multi-view attention of the latter 6 layers of the model is trained (total 12 layers). We run evaluations with 3 different seeds and average the results for experiments. Unless otherwise stated models are trained with the *Ensembled* dataset. For classification, we use do nearest neighbor search of the multi-view shape embedding with class embeddings. As in OpenShape, for each class label, we construct several sentences with the label and then take the average of the sentence embeddings as the class embedding. The final class chosen is the class with the highest average similarity between the class embedding and the shape embedding.

We compare the performance of our model against two CLIP models for different numbers of frames in Fig. 5, and recent state-of-the-art models in Tab. 2. For the CLIP models, we selected VIT with

Table 2: Classification accuracy on Objaverse-LVIS (O-LVIS) and ScanObjectNN trained on *Ensembled (no LVIS)* and *Ensembled*. As baselines, the zero-shot CLIP models (ZS B-32 and ZS BigG-14) are not trained on 3D data, whereas the fine-tuned CLIP model (FT B-32) is trained using a single image per object. For each model, we indicate the representation used for computing the embeddings, multi-view (MV) or point-cloud (PC), and the shape encoder (Enc) architecture that is used. Our multi-view based Duoduo CLIP is able to outperform point-cloud based approaches with the exception of Uni3D is a large model with 1B parameters. † indicates our reimplementation of OpenShape which is faster and outperforms the original OpenShape. F represents the number of frames/multi-views used for evaluation.

| Pretrain Dataset | | | Ensembled (no LVIS) | | | Ensembled | | | | | |
| | | | O-LVIS | | | O-LVIS | | | ScanObjectNN | | |
| Method | Rep | Enc | Top1 | Top3 | Top5 | Top1 | Top3 | Top5 | Top1 | Top3 | Top5 |
|---|---|---|---|---|---|---|---|---|---|---|---|
| ZS B-32 (12F) | MV | Avg | 35.7 | 54.8 | 62.1 | 35.7 | 54.8 | 62.1 | 53.9 | 73.5 | 81.2 |
| ZS BigG-14 (12F) | MV | Avg | 43.5 | 64.2 | 71.3 | 43.5 | 64.2 | 71.3 | 56.7 | 78.2 | 85.8 |
| FT B-32 (12F) | MV | Avg | 50.1 | 72.0 | 79.2 | 53.0 | 74.7 | 81.4 | 55.1 | 75.6 | 83.9 |
| ULIP (Xue et al., 2023) | PC | PointBERT | 21.4 | 38.1 | 46.0 | 26.8 | 44.8 | 52.6 | 51.6 | 72.5 | 82.3 |
| OpenShape (Liu et al., 2023a) | PC | PointBERT | 39.1 | 60.8 | 68.9 | 46.8 | 69.1 | 77.0 | 52.2 | 79.7 | 88.7 |
| OpenShape† | PC | PointBERT | – | – | – | 50.0 | 72.0 | 79.2 | - | - | - |
| TAMM (Zhang et al., 2024) | PC | PointBERT | 42.0 | 63.6 | 71.7 | 50.7 | 73.2 | 80.6 | 55.7 | 80.7 | 88.9 |
| MixCon3D Gao et al. (2024) | PC + MV | PointBERT | 47.5 | 69.0 | 76.2 | 52.5 | 74.5 | 81.2 | 58.6 | 80.3 | 89.2 |
| Uni3D (Zhou et al., 2024) | PC | 3D VIT | 47.2 | 68.8 | 76.1 | **55.3** | 76.7 | 82.9 | 65.3 | 85.5 | **92.7** |
| ShapeLLM (Qi et al., 2024) | PC | RECON++ | – | – | – | 53.7 | 75.8 | 82.0 | 65.4 | 84.1 | 89.7 |
| VIT-LENS (Lei et al., 2024) | PC | VIT-LENS$_G$ | 50.1 | 71.3 | 78.1 | 52.0 | 73.3 | 79.9 | 60.1 | 81.0 | 90.3 |
| Duoduo CLIP (5F) | MV | MVA | 51.3 | 73.1 | 79.9 | 54.1 | 76.0 | 82.4 | 60.7 | 82.4 | 88.5 |
| Duoduo CLIP (12F) | MV | MVA | **52.7** | **74.5** | **81.3** | 55.2 | **77.1** | **83.4** | **66.3** | **85.5** | 90.2 |

B-32 (the model that our Duoduo CLIP is initialized with), and BigG-14 (the CLIP teacher model used by OpenShape). Both models are run in a zero-shot manner without training on additional 3D data (the weights are frozen, with embeddings from the image encoder averaged). We also include an additional baseline that mirrors our default model's settings but is trained by sampling a single view per object. This setup simulates the single-image training of the original CLIP model, and we refer to it as FT B-32 in the tables.

**How well does performance scale with number of views?** Having a more complete view of a 3D object is important for 3D representation learning as parts of the object may be occluded for some views. Thus we investigate how the number of views affects the classification accuracy. From Fig. 5 (left), we see that our model performance scales better as the amount of information (views) increases compared to the zero-shot CLIP models (11.22 for our Duoduo CLIP, and 7.45 for B-32, the model we used for initializing our weights). This suggests that our training strategy of randomly sampling 1 to 6 views during training is effective in letting the model learn to leverage the number of views given to it. Although, the accuracy increases as the number of multi-views increases, the inference time also increases (Fig. 5 right). As the accuracy tapers off as we increase the number of views, to avoid additional computational cost during inference, we use 12 views as the default.

**How many multi-views to beat a point cloud?** Compared with using multi-views, using a point cloud representation offers good coverage of the entire geometry of the 3D object. Thus, we ask the question of how many views does it take to match existing methods on the Objaverse-LVIS classification task. We compare against ULIP (Xue et al., 2023), OpenShape (Liu et al., 2023a), TAMM (Zhang et al., 2024) that use point clouds with PointBERT (Yu et al., 2022b) as the 3D encoder, but different ways of fine-tuning the 3D encoder. We also compare against more recent methods (Lei et al., 2024; Qi et al., 2024; Zhou et al., 2024) that are the current top-performing methods on Objaverse-LVIS. From Tab. 2, on the *Ensembled* dataset, we are able to beat most previous methods with 5 multi-views except for Uni3D which has a much larger model (1B parameters). When we increase to 12 frames we are on par and even surpass Uni3D. This is also while using much fewer resources as seen in Tab. 1.

**Does multi-views generalize better compared to point cloud?** From the *Ensembled (no LVIS)* column of Tab. 2, we see that even the zero-shot BigG-14 model used as the teacher model for OpenShape beats it. Suggesting that the CLIP models already generalize well to unseen data, and point cloud methods potentially having worse generalization on unseen shapes. With our model that is further trained on *Ensembled (no LVIS)* we surpass point cloud methods by a large margin. Showing that fine-tuning the CLIP model better preserves their generalization ability. Additionally, our model generalizes well on ScanObjectNN (Uy et al., 2019) and outperforms point cloud methods on both Top1 and Top3 accuracies (see App. A.3 for more results on ScanObjectNN and ScanNet).

Table 3: MVPNet classification comparison.

| Method | Top 1 | Top 3 | Top 5 |
|---|---|---|---|
| ZeroShot B-32 (12F) | 52.68 | 70.99 | 77.22 |
| FT B-32 (12F) | 44.43 | 63.12 | 70.24 |
| OpenShape† | 10.80 | 19.62 | 25.20 |
| OpenShape† (+MVPNet) | 54.59 | 72.66 | 78.61 |
| Ours (12F) | 49.16 | 66.96 | 74.12 |
| Ours (+MVPNet) (1F) | 59.23 | 76.12 | 81.74 |
| Ours (+MVPNet) (6F) | 64.18 | 80.80 | 85.92 |
| Ours (+MVPNet) (12F) | 64.44 | 81.11 | 85.97 |
| Ours (+MVImgNet) (1F) | 61.75 | 79.07 | 84.18 |
| Ours (+MVImgNet) (6F) | 65.70 | 82.44 | 87.08 |
| Ours (+MVImgNet) (12F) | **66.06** | **82.72** | **87.21** |

Table 4: Text to shape retrieval comparison. Methods are trained on the Text2Shape Chen et al. (2019) dataset (T2S) or *Ensembled*.

| Model | Data | RR@1 | RR@5 |
|---|---|---|---|
| Text2Shape Chen et al. (2019) | T2S | 0.40 | 2.37 |
| Y2Seq2Seq Han et al. (2019) | T2S | 2.93 | 9.23 |
| Parts2Words Tang et al. (2023) | T2S | 12.72 | 32.98 |
| TriCoLo Ruan et al. (2024) | T2S | 12.22 | 32.23 |
| MXM-CLR Wang et al. (2023) | T2S | 16.83 | 39.06 |
| ZeroShot B-32 | Images | 11.90 | 27.85 |
| FT B-32 | Images | 15.01 | 32.46 |
| OpenShape† | Ens. | 10.53 | 25.94 |
| Uni3D-G | Ens. | 10.78 | 26.37 |
| Duoduo CLIP | Ens. | **15.90** | **34.08** |

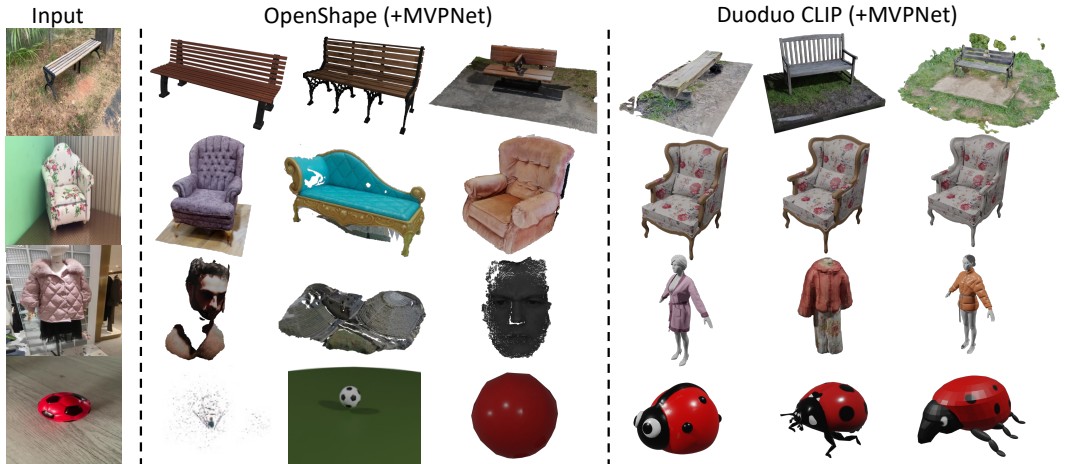

Figure 6: Retrieval results for OpenShape and Duoduo CLIP given input images from MVPNet. We perform retrieval with 12 input images, on the leftmost column we show one of the frames of the query object. For both methods we show the top 3 retrieval results using Objaverse as a retrieval library. Overall, Duoduo CLIP retrieves shapes that are more similar to the query image, while OpenShape sometimes retrieves broken geometry (row 3).

### 4.2.2 MVImgNet

To investigate how well our Duoduo CLIP works on real-world images, we evaluate on multi-view images from MVImageNet. For these experiments, we take our Duoduo CLIP model trained on the *Ensembled* dataset and add additional training data from MVImgNet. We consider two sets of data from MVImgNet, the MVPNet training split of objects or the entire MVImgNet (after our preprocessing and filter). For comparison with point cloud based methods, we evaluate on the validation set of MVPNet. As a representative of point cloud based methods, we choose OpenShape as it is relatively resource-light compared to other point cloud methods.

**Classification results.** We report accuracy results on MVPNet in Tab. 3. For OpenShape, the top-1 accuracy is very low (only $10.8\%$ top-1) when evaluating on MVPNet directly with the model trained on *Ensembled* only. We suspect this is due to noisy reconstructed point clouds and inconsistent orientations compared to the training data. Note we re-align MVPNet point cloud's up direction with Objaverse. After training on MVPNet data, OpenShape's performance improves significantly. For our Duoduo CLIP, even without training on MVImgNet or MVPNet data, our model achieves a top-1 accuracy of $49.16\%$. However, this is lower than the zero-shot CLIP B-32 model that our model is initialized from, suggesting possible overfitting to synthetic data. When we include training with multi-view images from MVPNet, our method surpasses both OpenShape and the zero-shot CLIP B-32 model even when just using one 1 view. Performance is further boosted when trained on the entire MVImgNet, highlighting the advantages of our Duoduo CLIP which uses multi-view images over OpenShape's point clouds. By using multi-view images, we have better generalization

on unseen data due to the priors from initializing with CLIP B-32. We also can train on more data, as multi-view images are more readily available compared to point clouds.

**Retrieval examples.** In Fig. 6, we present example retrieval results for querying Objaverse objects using images from the MVPNet validation set. We compare retrievals using OpenShape (+MVPNet) and Duoduo CLIP (+MVPNet). For OpenShape, we use 12 query images for each object, encode them with the frozen BigG-14 CLIP model, and average the image embeddings to obtain the query embedding.[1]. This embedding is then matched against a database of point cloud embeddings of Objaverse objects to retrieve the top 3 results. For our method, Duoduo CLIP encodes both the query and Objaverse objects using 12 multi-view images.

From Fig. 6, we see that for basic objects like benches, both methods are able to retrieve plausible objects. For the armchair (row 2), while OpenShape retrieved objects of the same semantic class, our Duoduo CLIP retrieved armchairs that has better matching textures. This is because multi-view images can represent textures better than point clouds. Note that some retrieved objects look similar due to duplicate objects in Objaverse. For the jacket (row 3), the objects returned by OpenShape have broken geometry and are incoherent. We hypothesize this may happen when the sampled points produce embeddings that are by chance similar to the query image. This issue is also present last example (a red and black soccer ball) where the first object retrieved by OpenShape has broken geometry. However, our method also fails in this case, as our model retrieves ladybugs with red wings and black dots that resemble the soccer ball. Interestingly, OpenShape is able to retrieve a soccer ball and a red ball. This can indicate that point cloud based methods may be more robust to geometric properties of the object compared to the rendered images. However, we find that overall the retrieved objects from our method are more consistent.

### 4.2.3   MODEL ABLATIONS

Table 5: Ablation on the number of layers for accuracy on Objaverse-LVIS using 12 input frames. Default model highlighted in gray.

| Method | Top 1 | Top 3 | Top 5 |
|---|---|---|---|
| 3 layers | 53.77 | 75.8 | 82.41 |
| 6 layers | 55.24 | 77.08 | 83.43 |
| 12 layers (full) | **55.32** | **77.08** | **83.49** |

Table 6: Multi-view attention (MVA) ablation of accuracy on Objaverse-LVIS (O-LVIS), MVPNet and ScanObjectNN with 12 frames. Default model highlighted in gray.

| Method | Layers | O-LVIS | MVPNet | ScanObjectNN |
|---|---|---|---|---|
| -MVA | 6 | 54.61 | 47.87 | 58.77 |
| +MVA | 6 | 55.24 | **49.16** | **66.32** |
| -MVA | 12 | 55.02 | 43.75 | 56.83 |
| +MVA | 12 | **55.32** | 44.42 | 64.15 |

**Do we need to train all the layers?** We conduct an ablation to see if it is necessary to train all 12 layers. In Tab. 5, we show the accuracy for restricting the training to the last 3 and 6 MVA layers vs the full 12 MVA layers. We see that the number of trainable layers can be cut by half with minimal losses to accuracy, while further cuts start to hurt model performance. Training only half the layers reduces the VRAM usage by roughly half and allows us to train on GPUs with less memory. By restricting the training to the last 6 MVA layers, we can shift from training on $4 \times$A40s (48 GB) to $4 \times$A5000s (24 GB) (see Tab. 1). Also we find that training less layers also helps preserve the priors of the model and make generalization on other datasets better as seen in Tab. 6.

**Does multi-view attention help?** We verify the effectiveness of our cross view attention in Tab. 6. +MVA and -MVA denote whether the cross-views attention is enabled. On Objaverse-LVIS, while the +MVA mechanisms help less **+0.3** when all 12 layers are trainable, it makes a larger impact when only training 6 layers **+0.63**. Making the +MVA mechanism an effective way to maintain performance while being more efficient compared to training all 12 layers. In addition, we find that the +MVA also helps in making the model generalize better on unseen datasets like MVPNet and ScanObjectNN compared to -MVA. Inference speeds for the models can be found in App. A.1.1.

### 4.3   FINE GRAINED TEXT-TO-SHAPE RETRIEVAL

Previous evaluations focused on classification or image-based retrieval and do not assess how well the methods align fine-grained text descriptions with 3D models. To evaluate this aspect, we use the test

---

[1]For the largest BigG-14 CLIP model, OpenShape does not use any linear projection layer $g^I$

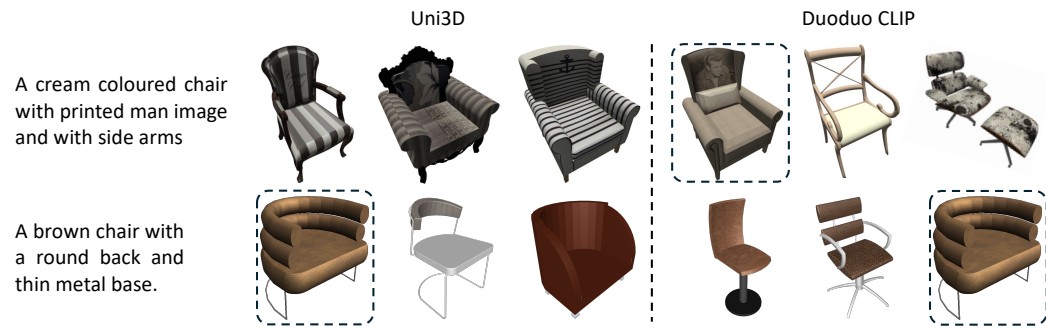

Figure 7: Retrieval using fine-grained text from the Text2Shape dataset. On the left is the input text description and we show the top 3 retrieval results for Uni3D and Duoduo CLIP with the ground truth shape highlighted in dotted boxes.

split of Text2Shape (Chen et al., 2019), which contains ∼1.4k tables and chairs from ShapeNet and ∼7.4k human-captioned descriptions (∼5 per shape). We report the shape retrieval rates at 1 and 5.

**Retrieval performance.** We compare the performance of our model to prior work in Tab. 4. We also include previous state-of-the-art methods that were trained on the Text2Shape dataset. Overall, compared to other methods trained on the *Ensembled* dataset, our method demonstrates superior performance. The zero-shot performance alone is impressive, surpassing point cloud methods. Notably, we observed a significant discrepancy between the zero-shot CLIP model reported by Ruan et al. (2024) and our findings, where the LAION checkpoint trained by OpenCLIP significantly outperforms the original OpenAI checkpoint. Lastly, our model approaches the performance of the SOTA method trained directly on Text2Shape, indicating that there is still much to explore regarding more fine-grained text alignment in large-scale text-to-3D models.

**Retrieval examples.** We present example retrieval results compared to Uni3D in Fig. 7. Generally, for text descriptions emphasizing complex textures, such as the 'man image' in the first row, our model successfully retrieves the corresponding ground truth 3D model. However, for descriptions focusing on geometric properties, like 'round back' in the second row, point cloud-based methods tend to retrieve shapes that better match the specified criteria.

## 5 CONCLUSION

We introduced Duoduo CLIP, which uses a multi-view image encoder to represent shapes and align the embedding to pretrained text-image models. We demonstrated that contrastive learning with multi-view images is an efficient and effective way to encode 3D shapes. Our experiments show that using multi-view images our model surpasses current point cloud state-of-the-art methods, trained in a similar fashion, on both real and synthetic data while using significantly fewer resources.

Multi-view images have limitations. They are potentially poorer at geometric understanding compared to point clouds. One way to alleviate this is to incorporate additional inputs such as depth or normal images, which provide more geometric cues. Our work is also limited in only exploring CLIP-style contrastive pretraining. Our model can be easily adapted to other contrastive pretraining Goel et al. (2022); Zhai et al. (2023) and SSL methods (Oquab et al., 2023; Zhou et al., 2022). This is a promising direction as recent work (Banani et al., 2024) points out the relative weakness of CLIP pretraining on both 3D structure and consistency benchmarks. Another important direction is to examine how to increase the data available for training. Even with a relatively small model as ours, training too many layers can lead to severe overfitting. This suggests that data augmentation and incorporating more 3D data would lead to improved generalization and performance.

By advocating for a reconsideration of multi-view images for shape representation, we believe our work can convince 3D shape researchers to re-examine when point clouds are useful. Images offer several advantages over point clouds, including better capture of texture and color information, less domain gap with pretrained vision models, and better resource scaling. They also offer greater flexibility for practical use cases in real-world applications such as robotics. Leveraging the texture information captured by multi-view methods like ours can complement the geometric understanding of point cloud techniques. A hybrid approach combining both modalities could leverage their respective strengths and adapt to a variety of use cases in the future.

**Acknowledgements.** This work was funded by a CIFAR AI Chair, an NSERC Discovery grant, and a CFI/BCKDF JELF grant.

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

# A    APPENDIX

In this appendix, we provide details of inference speed (A.1.1), camera sampling (A.1.2), as well as ablations of the number of multi-views used during training (A.2.1), MLP fine-tuning (A.2.2), and different initialization of encoders (A.2.3). We also include experiments on real-world scanned datasets (A.3), examples of out-of-distribution retrieval (A.4.1), concept mixing (A.4.2), and details of the attention map visualization (A.5.1).

For the ablation experiments in the appendix unless otherwise mentioned, we follow the same default settings as in the main paper. In this default setting, models are trained on the Ensembled dataset and we randomly sample 1 to 6 multi-views during training, with the last 6 attention layers trainable and +MVA enabled using a batch size of 1600.

## A.1    INFERENCE SPEED AND CAMERA SAMPLING

### A.1.1    INFERENCE SPEED

We show the inference speed between different number of trainable layers and frames for -MVA and +MVA in Tab. 7. We benchmark total run time and time per iteration on the entire Objaverse-LVIS (46k shapes). Note that -MVA with 6 layers is not shown, since the run time is the same with -MVA 12 layers during inference since both models have multi-view attention disabled. There is only a slight increase in inference time when +MVA is enabled, and it remains efficient for processing a large number of shapes.

Table 7: Inference speed on Objaverse-LVIS with batch size 200. Default model is highlighted in gray.

| Method | Layers | Frames | Speed (s/it) | Total Time (s) |
|--------|--------|--------|--------------|----------------|
| +MVA | 6 | 3 | 0.14 | 32.80 |
| +MVA | 6 | 6 | 0.28 | 65.52 |
| +MVA | 6 | 12 | 0.56 | 130.67 |
| -MVA | 12 | 3 | 0.14 | 33.23 |
| -MVA | 12 | 6 | 0.28 | 63.90 |
| -MVA | 12 | 12 | 0.54 | 126.28 |
| +MVA | 12 | 3 | 0.15 | 34.01 |
| +MVA | 12 | 6 | 0.29 | 68.38 |
| +MVA | 12 | 12 | 0.59 | 136.10 |

Table 8: Different camera sampling settings. Here the elevation is the angle relative to the horizon, with 0 degrees at the horizon, positive angles looking from upward, and negative angles looking from downward.

| Setting | Radius | Azimuth | Elevation |
|---------|--------|---------|-----------|
| UpperHem | [1.5, 2.2] | [0, 360] | [0, 60] |
| ExtendedHem | [1.5, 2.2] | [0, 360] | [-30, 60] |
| FullSphere (Zero123) | [1.5, 2.2] | [0, 360] | [-90, 90] |

### A.1.2    CAMERA SAMPLING

To verify the effectiveness of our camera sampling strategy, we benchmark different camera settings (see Tab. 8). Example renders for each of the different settings as well as the quantitative results can be found in Fig. 8. The results show a clear performance improvement as the elevation range increases, suggesting that capturing the object from more extreme angles also provides more context and information about the shape.

## A.2    OTHER ABLATIONS

### A.2.1    NUMBER OF MULTI-VIEWS DURING TRAINING

For our default model we choose to sample 1 to 6 multi-views during training as it strikes a good balance between memory usage and performance. In Tab. 9 we show the Objaverse-LVIS accuracy over a variation of sampled views seen during training with other settings kept the same as the default model. Except for the model that is trained with only a single view ie. [1, 1] which uses -MVA as it does not see any multi-views during training.

It can be seen that as the number of multi-views seen during training increases, so does its performance when evaluating at 12 frames during inference. However, the memory usage also rises quickly as attention has quadratic growth when the the number of tokens increases in the case of +MVA. So we

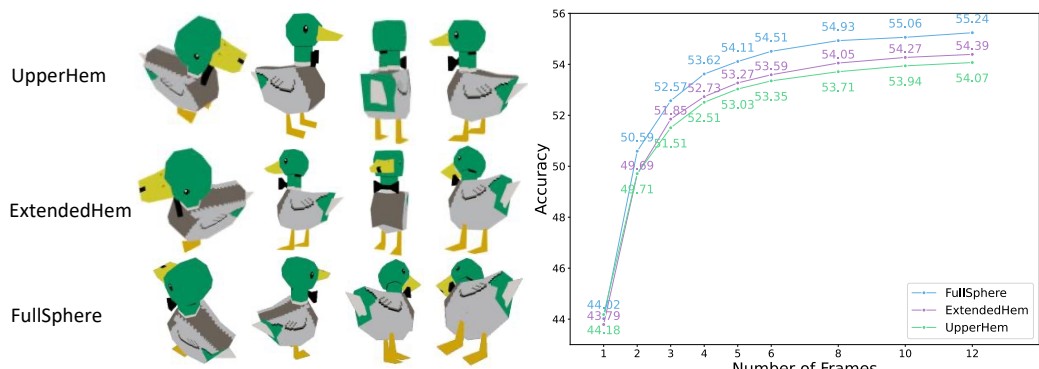

Figure 8: **Left.** We show example renders from each of the different camera settings. **Right.** We show the Top 1 accuracies of the different camera settings using our default model +MVA (6 layers).

choose to sample [1, 6] views as there is an increase of **+0.24** on top1 over training with [1, 4] views. However, increasing further to [1, 8] frames only results in a **+0.09** top1. Also the amount of memory [1, 6] also allows us to stay within the memory limits of an A5000 (24 GB) GPU.

Table 9: Number of views sampled during training on Objaverse-LVIS ablation. Evaluated using 12 frames per shape during validation. The memory usage is reported on a single GPU with 400 batch size. Default model is highlighted in gray.

| Num Frames Train | Top1 | Top3 | Top5 | Mem (GB) |
|---|---|---|---|---|
| [1, 1] | 52.97 | 74.71 | 81.44 | 5.14 |
| [1, 2] | 54.47 | 76.37 | 82.82 | 9.12 |
| [1, 4] | 55.00 | 76.93 | 83.36 | 15.50 |
| [1, 6] | 55.24 | 77.08 | 83.43 | 22.18 |
| [1, 8] | 55.33 | 77.30 | 83.62 | 28.92 |

Table 10: Ablation of also training the MLP layers after attention on Objaverse-LVIS, MVPNet and ScanObjectNN. Evaluated using 12 frames of each shape during validation. Default model is highlighted in gray. ‡ Trained with batch size 1400 to fit in memory.

| Method | Layers | O-LVIS | MVPNet | ScanObjectNN |
|---|---|---|---|---|
| +MVA | 6 | 55.24 | 49.16 | 66.32 |
| +MVA | 12 | 55.32 | 44.42 | 64.15 |
| +MVA + MLP | 6 | 55.96 | 45.46 | 59.63 |
| +MVA + MLP ‡ | 12 | 55.63 | 32.68 | 60.20 |

### A.2.2 MLP FINE-TUNING

Only fine-tuning the attention layers of the model not only makes the model more efficient to train, but also helps to preserve the priors of the original CLIP model. To showcase this we also enable the training of the MLP layers corresponding to the attention layers being tuned in Tab. 10. Although we find that Objaverse-LVIS performance improves when more of the network (MLP) is allowed to be tuned. It results in worse performance on the other unseen datasets. Our default model performs the best across the other unseen datasets. Note that although our default model also overfits and performs worse on MVPNet compared to the zero-shot model as seen in Tab. 3, the performance drop-off is smaller compared to the other models.

### A.2.3 DIFFERENT INITIALIZATION

To understand the importance of initializing the multi-view shape encoder from the same model as the frozen CLIP encoders we conduct additional experiments by using a different initialization. Namely, we use the same ViT-B/32 architecture, but initialize the shape encoder from the OpenAI (Radford et al., 2021) pretrained model and the frozen CLIP model uses the checkpoint *laion2b_s34b_b79k* from the open source implementation OpenCLIP (Ilharco et al., 2021). Note that we add an additional linear layer for the embeddings of the OpenAI initialized model. The results can be found in Tab. 11.

From the results, we can see that using a different initialization from the frozen clip model results in a large drop-off in performance for both Objaverse-LVIS and ScanObjectNN. This shows the importance of using the same initialization for both shape and frozen CLIP encoders. As the embedding space outputted by the models are already close and allows for smoother fine-tuning and prevents potential "overfitting" to the frozen target text and image embeddings. However, we expect

that this problem will diminish with larger datasets, allowing for the use of even more powerful 2D foundational models for the shape encoder such as the DINO models (Caron et al., 2021; Oquab et al., 2023) that also provide better 3D Awareness (Banani et al., 2024).

Table 11: Results of using different initializations for the shape encoder. Same Init follows the settings of the main paper which uses the OpenCLIP pretrained model for both the shape and frozen CLIP encoders. OpenAI Init uses the pretrained model from OpenAI for the shape encoder instead. We evaluate on Objaverse-LVIS and ScanObjectNN with 12 views. We show results for models with different number of trainable layers.

| Method | Layers | O-LVIS | | | ScanObjectNN | | |
|---|---|---|---|---|---|---|---|
| | | Top1 | Top3 | Top5 | Top1 | Top3 | Top5 |
| OpenAI Init | 6 | 53.0 | 75.2 | 82.1 | 60.8 | 78.6 | 86.6 |
| OpenAI Init | 12 | 53.4 | 75.7 | 82.5 | 56.8 | 78.9 | 86.8 |
| Same Init | 6 | 55.2 | 77.1 | 83.4 | 66.3 | 85.5 | 90.2 |
| Same Init | 12 | 55.3 | 77.1 | 83.5 | 64.2 | 85.0 | 90.8 |

## A.3 OTHER DATASETS

### A.3.1 SCANOBJECTNN

To evaluate on ScanObjectNN (a point cloud dataset), we have to first obtain the multi-view images for each object. For simplicity, we rendered the 12 multi-view images using the point clouds (OBJ_BG) directly as can be seen in Fig. 9. Note that ScanObjectNN objects were obtained from real-world scan datasets that were captured with RGBD cameras. So using the raw frames would be more straightforward and easier to obtain compared to point clouds since it does not require reconstruction. However, we weren't able to obtain the mappings to the original scenes from the processed ScanObjectNN dataset so opted for this approach instead.

We show the quantitative results on ScanObjectNN in Tab. 12. It can be seen that the +MVA mechanism outperforms -MVA by a large margin in this scenario. And that our default model with 12 frames are able to outperform all previous point cloud methods on both top 1 and top 3 accuracies. This is despite the fact that we used rendered point clouds images as opposed to the originally captured RGB scene data. This demonstrates the robustness and adaptability of our model to handle a wide range of different images.

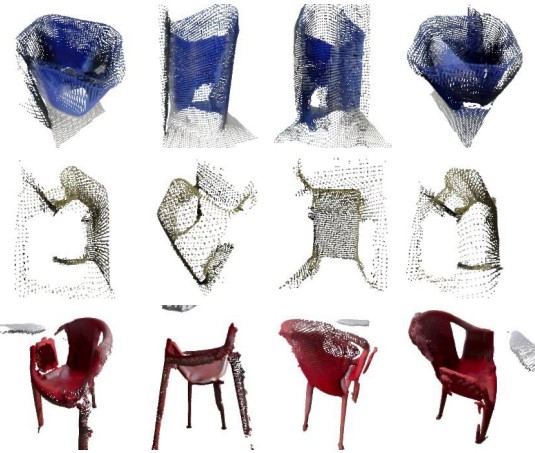

Figure 9: Example of multi-view image renders of ScanObjectNN.

Table 12: ScanObjectNN Results. Our +MVA model significantly outperforms zero-shot CLIP and -MVA. We also outperforms prior work using point clouds, including Uni3D on Top 1 accuracy.

| Method | Frames | Top 1 | Top 3 | Top 5 |
|---|---|---|---|---|
| ZS B-32 | 1 | 37.2 | 60.3 | 71.6 |
| ZS B-32 | 6 | 50.9 | 72.4 | 81.2 |
| ZS B-32 | 12 | 53.9 | 73.5 | 81.2 |
| -MVA | 1 | 41.2 | 62.8 | 74.4 |
| -MVA | 6 | 53.7 | 75.9 | 84.4 |
| -MVA | 12 | 56.8 | 78.4 | 85.5 |
| OpenShape | PC | 52.2 | 79.7 | 88.7 |
| Recon++ | PC | 65.4 | 84.1 | 89.7 |
| Uni3D | PC | 65.3 | 85.5 | **92.7** |
| +MVA (6 layers) | 1 | 41.7 | 64.4 | 76.4 |
| +MVA (6 layers) | 6 | 61.7 | 82.7 | 89.2 |
| +MVA (6 layers) | 12 | **66.3** | **85.5** | 90.2 |

### A.3.2 SCANNET

For ScanNet, we generate the multi-view images using the officially released projections from the 3D segmentations onto the original 2D RGB frames. We then sort the frames based on the projected

Table 13: ScanNet results with cropped video frames evaluated over 12 multi-views. ZS is the zero-shot B-32 model. Ens and Ens + MVI is the default model trained with ensembled dataset and ensembled with MVImgNet datasets respectively.

| Method | Avg. | Bed | Cab | Chair | Sofa | Tabl | Door | Wind | Bksf | Pic | Cntr | Desk | Curt | Fridg | Bath | Showr | Toil | Sink |
|---|---|---|---|---|---|---|---|---|---|---|---|---|---|---|---|---|---|---|
| OpenShape | 45.6 | 66.7 | 3.2 | 75.8 | 83.5 | 37.7 | 49.2 | 47.5 | 64.9 | 48.2 | 1.9 | 66.1 | 70.2 | 1.8 | 50.0 | 57.1 | 45.2 | 7.1 |
| TAMM | 49.4 | 66.7 | 4.8 | **83.6** | 84.5 | 48.9 | 57.9 | 48.2 | 80.5 | 61.3 | 1.9 | 60.6 | 83.6 | 7.0 | 41.4 | 56.1 | 48.4 | 3.6 |
| Uni3D | 45.8 | 58.5 | 3.7 | 78.8 | 83.7 | 54.9 | 31.3 | 39.4 | 70.1 | 35.1 | 1.9 | 27.3 | **94.2** | 13.8 | 38.7 | 10.7 | 88.1 | 47.6 |
| ZS | **69.2** | **83.1** | **58.2** | 65.9 | **90.3** | **56.3** | 56.9 | 43.5 | 86.2 | **67.7** | **44.2** | **93.7** | 47.0 | 79.0 | **64.5** | 62.0 | 99.4 | 78.6 |
| Ens | 61.3 | 83.1 | 50.0 | 46.1 | 87.2 | 40.5 | 53.4 | 39.6 | 97.0 | 45.9 | 16.0 | 89.2 | 65.7 | **86.6** | 33.3 | 56.0 | 98.0 | 55.1 |
| Ens+MVI | 63.3 | 87.2 | 51.3 | 56.5 | 88.5 | 35.1 | **63.3** | 49.5 | **98.7** | 48.8 | 25.0 | 92.9 | 55.1 | 77.8 | 26.9 | 54.8 | 98.9 | 65.0 |

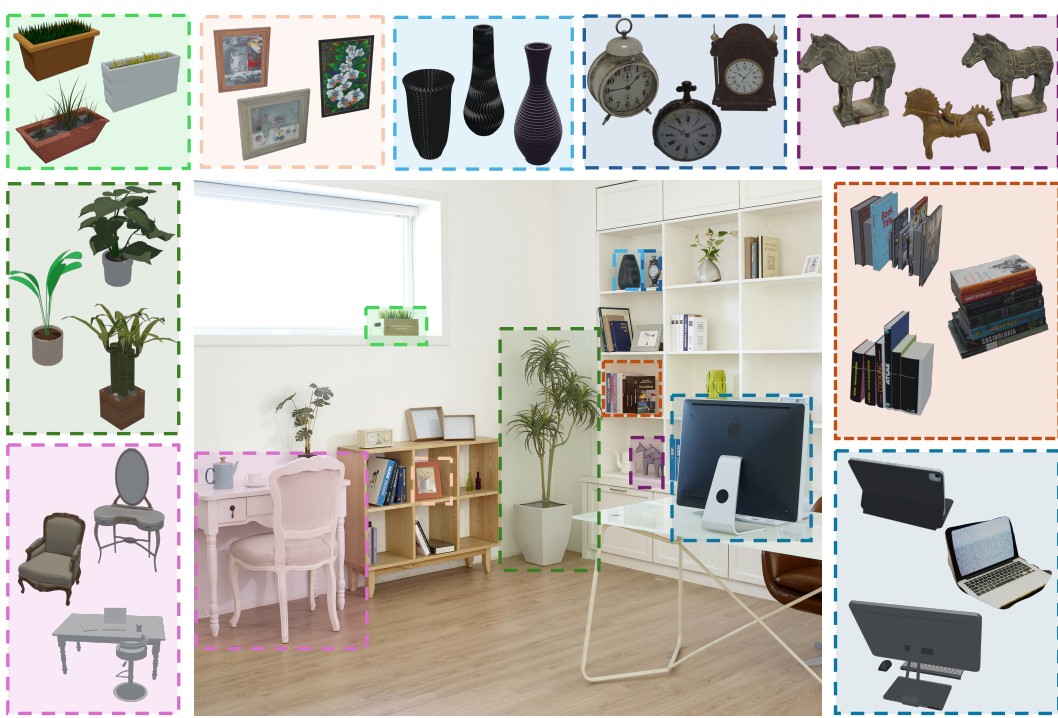

Figure 10: We perform out-of-distribution image-based retrieval on Objaverse LVIS by using manually cropped images from a large indoor scene image sourced from the web to retrieve the top 3 matching objects using our DuoduoCLIP model trained with MVImgNet.

object mask area, selecting the 12 views with the largest areas. Afterward, we crop each object according to its bounding box, centering it within the frame and filling the regions outside the bounding box with white. We compare against recent point cloud based methods in Tab. 13.

It is evident that image-based models significantly outperform point cloud models. Notably, the best-performing model is the zero-shot B-32 CLIP model, highlighting a persistent issue with dataset size and the tendency of these methods to overfit. While we observe improvements from models trained on both Ensembled and MVImgNet datasets, which include natural images, the gains are still small, suggesting that larger datasets could lead to further performance enhancements.

## A.4 ADDITIONAL RESULTS

### A.4.1 OUT OF DISTRIBUTION RETRIEVAL

To further assess our model's understanding of 3D objects, we conduct image-based retrieval using out-of-distribution images. We source indoor scene images from the web and generate additional images using a large text-to-image model (Esser et al., 2024). Objects are manually cropped from these scenes (1 view each) and processed through our shape encoder to obtain query embeddings. For the retrieval database, we compute embeddings for the Objaverse LVIS split using 12 views per

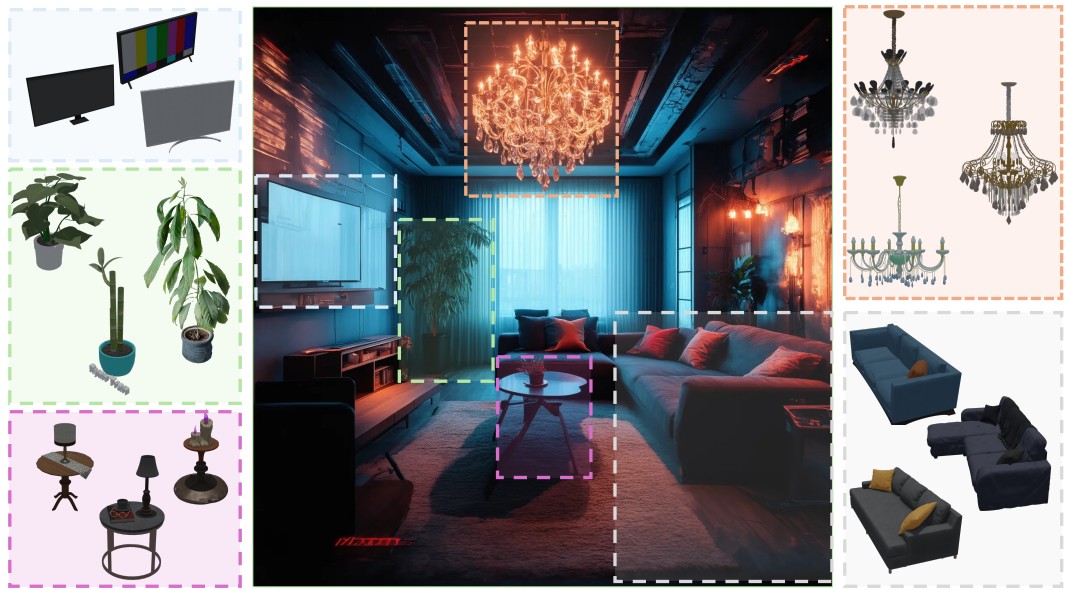

Figure 11: We perform out-of-distribution image-based retrieval on Objaverse LVIS by using manually cropped images from a large indoor scene image generated from Stable Diffusion 3 (Esser et al., 2024) to retrieve the top 3 matching objects using our DuoduoCLIP model trained with MVImgNet.

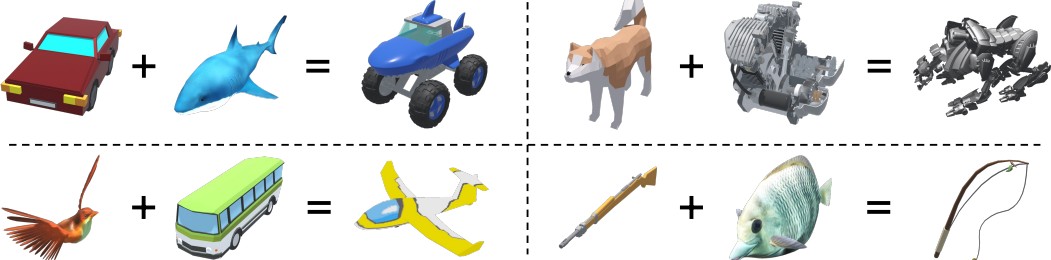

Figure 12: Additional examples of context mixing where we blend concepts from two distinct objects by identifying the Objaverse shape with the embedding that maximizes similarity to both objects.

object. The top 3 matches are returned based on the highest cosine similarity between the query embeddings and the database embeddings. The results for the image sourced from the web are shown in Fig. 10, while those for the image generated by the text-to-image model are presented in Fig. 11.

Our model successfully retrieves objects that are semantically related to the corresponding cropped portions of the images, demonstrating a strong understanding of object categories and their semantic significance. While the retrieved objects may not visually match the cropped images exactly, it is important to note that these are in-the-wild images, and identical objects may not exist in the Objaverse LVIS dataset.

### A.4.2 CONCEPT MIXING

To explore the latent space learned by our model, we perform concept mixing by identifying the object embedding within Objaverse that maximizes the similarity to both of the two objects being combined. Similar to OpenShape (Liu et al., 2023a), to blend shape $a$ with shape $b$, we select the object $\hat{i}$ with normalized shape embedding $f_{\hat{i}}^S$ such that $\hat{i} = \arg\max_i \left( \min(\langle f_i^S, f_a^S \rangle, \langle f_i^S, f_b^S \rangle) \right)$. Note that the shape embedding is extracted by encoding the 12 views of each object. The results are illustrated in Fig. 2 and Fig. 12. These examples demonstrate that our model has learned a meaningful latent space capable of blending geometric and semantic concepts. For instance, it can combine a pumpkin and a carriage into a pumpkin carriage (Fig. 2 right) or a horse and a wagon into a horse pulling a wagon (Fig. 2 left). Interestingly, the model can also combine semantic attributes, such

as merging a dog and an engine to create a four-legged mechanical creature (Fig. 12 top right) or blending a bird and a bus to form a plane (Fig. 12 bottom left), reflecting an understanding of the geometric function of wings and the concept of transport.

## A.5 ADDITIONAL DETAILS

### A.5.1 ATTENTION MAP VISUALIZATION

For Fig. 4, we visualize attention maps (Abnar & Zuidema, 2020) by first selecting an area of interest in one of the multi-view images (e.g., the plate in the first example). All 12 multi-view images of the object are passed through the model backbone to extract the attention map from the query and key embeddings in the first MVA layer. This produces an attention map of size $(M \times 49) \times (M \times 49)$, where $M$ is the number of multi-views and $49$ is the number of tokens per image (excluding the CLS token, with image resolution 224 and ViT patch size 32, giving $(224/32)^2 = 49$ patches). To retrieve attention specific to the area of interest (e.g., token 11 (plate) in the first image), we extract the corresponding row in the attention map, resulting in an attention response of size $M \times 49$. This response is overlaid onto the original images, as shown in Fig. 4. These visualizations reveal how the attention response to a token in one image corresponds to regions in other multi-view images, demonstrating the model's ability to capture geometric relationships across views.

