# OpenReview forum: "Duoduo CLIP: Efficient 3D Understanding with Multi-View Images"
_ICLR.cc/2025/Conference — ICLR 2025 Poster_

### Official Review · Reviewer_fQzu · 2024-10-30

**Soundness:** 3
**Presentation:** 3
**Contribution:** 2
**Rating:** 6
**Confidence:** 3

**Summary:**

The paper introduces a novel method named Duoduo Clip, which addresses limitations in 3D shape representation by using multi-view images over traditional point clouds, improving adaptability and efficiency in 3D shape retrieval. Unlike previous methods constrained by point cloud limitations and high computational demands (e.g., Uni3D), Duoduo CLIP leverages multi-view images to simplify training, generalize better to unseen shapes, and reduce the need for extensive compute resources. This approach allows direct input of images, enabling seamless integration with real-world datasets (e.g., ScanNet, ScanObjectNN) and enhancing real-time applicability in robotics.

**Strengths:**

Here is the strength of this paper:
- The authors propose to use rendered multiview images as shape representation instead of point cloud inputs for retrieval task.
- It proposes a training strategy that requires significantly less computation resources.
- The author claims SoTA performance in comparison with baseline models.

**Weaknesses:**

Here is some concerns from me:

**Major:**

- **Novelty:** While this paper proposes a more efficient approach to leverage multiview image for 3D understanding. Using multi-view (MV) attention to establish connections between multiple views for fine-tuning is not a novel idea. This limits the technical contribution. The primary innovation appears to be shifting from single-view images to multi-view images for training, which is incremental rather than groundbreaking. Also, this paper is similar with: `Yipeng Gao et al. Sculpting Holistic 3D Representation in Contrastive Language-Image-3D Pre-training CVPR2024`. Which use point cloud, multi-view image and text to conduct contrastive learning for multi-modality alignment. Author should explain about it.


- **Experiment.** I am also concerned about the potential degradation in performance on natural single-view images. The training dataset is smaller compared to the original CLIP model, and some of the data are synthetic. Although the proposed method improves performance on multi-view images, I worry that it may compromise the model’s ability to handle single-view natural images, which could negatively impact its performance in downstream applications.

**Minor:**

1.Line 80:
A citation is missing for the reference to "previous methods..."

2. There is some confusing in Figure 3:
- So why the first image all focus on the plate when multiple object are present? What is the query?
- I also question the claim that this figure demonstrates "geometric correspondences of 3D shapes." It could be interpreted as a simple classification heatmap, which does not necessarily indicate the model is aware of 3D structure.

**Questions:**

As posted in weakness, I hope the author could solve these concerns in the rebuttal phase.

---

> ### Author Response · Authors · 2024-11-21
> **Response to Reviewer fQzu**
>
> We thank the reviewer for reviewing our paper and highlighting the strengths of our work, including our use of rendered multiview images, efficient training strategy, and SoTA performance compared to baselines.
>
> ## Novelty
>
> Please see our comments on paper contributions and insights in the general response.
>
> Thank you for pointing out MixCon3D [Gao et al. CVPR 2024].  We have added a discussion of MixCon3D to our related work (Section 2) and Table 2.  Below we summarize the key differences between our work and MixCon3D.
>
> - MixCon3D fuses multi-view and point cloud features but trains only the point cloud encoder and aggregation layers, which does not leverage CLIP image encoder priors to improve multi-view embeddings. Their fused representation (point cloud + 4 views) achieves 52.5, 74.5, and 81.2 Top1, Top3, and Top5 accuracies on LVIS, while our model outperforms it (53.6, 75.3, 81.8) using only 4 multi-views.
> - MixCon3D relies on a fixed set of 12 predetermined views (4 sampled during training), limiting flexibility in handling arbitrary poses or view counts. Its multi-view features are aggregated from single-view CLIP embeddings without inter-view context within the CLIP backbone, unlike our MVA approach.
> - MixCon3D relies on point clouds and optional multi-view images, limiting its use in robotics scenarios with real-time, sparse views. In contrast, our model handles arbitrary views and poses, achieves better performance with only multi-view images, and is more resource-efficient, using 4 NVIDIA A5000 GPUs versus their 8.
>
> ## Experiment: Model’s ability to handle single-view natural images
>
> To maintain compatibility with arbitrary views, we train by sampling 1 to 6 multi-views, balancing single-view performance with multi-view adaptation. While training impacts zero-shot generalization on datasets like MVPNet (Table 3) when not fine-tuned on MVImgNet, this overfitting is less severe than in previous point cloud-based methods, which show lower zero-shot performance overall compared to ours. Also we further train our model on real-world multi-view images to improve generalization, unlike prior point cloud works that primarily train on synthetic datasets like Objaverse.
>
> ## Missing citation:
>
> Thanks for noticing this, we have added citations in the revision.
>
> ## Clarifying Figure 3 (visualization of attention maps):
>
> Sorry for the confusion regarding the generation of attention maps [1] in the paper. We have clarified this in the revised caption for Figure 3 and details in Appendix A.5.1. The attention map was extracted from the first trainable MVA layer, capturing token-wise attention between query and key embeddings. For 224x224 images with a patch size of 32, each image has 49 tokens (7x7 grid, excluding the CLS token), resulting in an attention map of size (M x 49) x (M x 49) across all M views. To visualize attention for the plate part in the first image, we identified its corresponding token (e.g., the 11th of 49) and retrieved its attention responses (size M x 49) across all views. These responses were overlaid on the original images to highlight attention magnitudes for the plate part, as shown in Figure 3, illustrating the model's learned geometric correspondences.
>
> [1] Abnar and Zuidema. “Quantifying Attention Flow in Transformers.” ACL 2020

---

> > ### Comment · Reviewer_fQzu · 2024-11-25
> >
> > Thank authors for addressing my concerns. The authors have answered most of my concerns, so I have decided to keep my original score.

---

> > > ### Author Response · Authors · 2024-11-29
> > >
> > > We appreciate your time and thoughtful feedback. Your suggestions have been helpful in improving the paper. We're pleased to know that our response have clarified your concerns.

---

### Official Review · Reviewer_p9H4 · 2024-11-04

**Soundness:** 3
**Presentation:** 3
**Contribution:** 3
**Rating:** 6
**Confidence:** 4

**Summary:**

This work presents Douduo CLIP, a technique for 3D representation learning for classification and retrieval that leverages multi-view input images rather than more explicit 3D representations like point clouds or meshes. The method is simple but effective approach to fine-tuning CLIP for 3D understanding, performing CLIP-style contrastive training between multi-view image encodings (a shape representation), single image encodings and text encodings. The multi-view image encoder is initialized from CLIP and fine-tuned with self-attention across all views for some layers of the ViT encoder, while the single image and text encoders are frozen (this allows for more efficient training).

The proposed Duoduo CLIP outperforms or is competitive with point cloud-based baselines for classification on Objaverse-LVIS and MVPNet, and outperforms CLIP applied zero-shot without any tuning. There are reasonable qualitative image-based retrieval and state-of-the-art zero-shot results (no specialized training for this task) for text-based retrieval.

**Strengths:**

### Originality
- It is reasonable to use pre-trained CLIP encoders and fine-tune them for 3D shape understanding from multiple-views. This allows for training on more data and taking advantage of the large-scale pre-trained representation.
- Attention implemented  across multiple views of a single object is reasonable, and ablation show that it helps.
### Quality
- The investigation into the effect of replacing the point-cloud encoder with a multi-view image encoder is thorough, including analysis about training/testing time inference costs, comparisons with non-fine tuned CLIP backbones, and comparisons with baselines on image-based retrieval, text-based retrieval and classification.
### Clarity
- The paper is mostly well written and easy to follow with the exception of the experiments section, which could be improved (see comments in weaknesses).
### Significance
- For tasks like retrieval and classification it is good to know that point-cloud representations are weaker than multi-view image representations, which is the main takeaway of this work. Point cloud representations find their main applications in robotics, often in object manipulation tasks that do not only require recognition and where real-time multi-view data collection is cumbersome (see comments further comments on significance in weaknesses)

**Weaknesses:**

### Originality
- The main technical novelty is replacing a point cloud encoder with a multi-view CLIP image encoder, and tuning this multi-view encoder. While effective, the novelty is minor. It would also be good to understand the significance of what pre-trained representation (CLIP vs. DINO for example) is used to initialize the multi-view encoder
### Quality
- Accuracy of evaluation: Both the Objaverse LVIS and the new MVPNet use somewhat automated procedures using off-the-shelf VLMs to create the object labels. Without additional filtering the Objaverse LVIS dataset is quite noisy and this is likely the case for BLIP (what was used for MVPNet) as well. This is also the case for techniques purpose-made for captioning like Luo et al. (NeurIPS 2023). What is the significance of the findings if the ground truth labels themselves are inaccurate and might contain similar mislabeling patters as the CLIP model that is used as the pre-trained representation in this work?
### Clarity
- The flow of the writing in the experiment section can be improved, the sentences don't flow into each other, but read as consecutive lists of facts (e.g.L301-314). It's possible to understand with some effort, but a revision would greatly improve the paper.
### Significance
- Point clouds are a data modality generally used for tasks that require fine-grained object shape understanding and real-time visual feedback in robotics, like object manipulation. The present work focuses on recognition related tasks, so the significance in robotics (emphasized by the authors) is not entirely clear.

**Questions:**

- Clarity: What is the difference between 5F and 12F as described in the tables (e.g. two bottom rows of Table 2)? This notation is not defined, although it's possible to conclude that this might mean number of views?
- What is the effect of the choice of pre-trained backbone for the multi-view encoder? How much worse would a DINO or randomly initialized model do? It would be good to understand the effect and significance of this choice.
- What are specific applications in robotics for which multi-view images are available in real time and where improved recognition from multi-view images would have a positive and significant impact?

---

> ### Author Response · Authors · 2024-11-21
> **Response to Reviewer p9H4**
>
> We thank the reviewer for taking time to read our paper. We're glad you found the use of fine-tuned CLIP encoders and attention across multiple views to be a reasonable approach, and we appreciate your recognition of the detailed analysis comparing point-cloud and multi-view image encoders. We also thank the reviewer for the insightful feedback for improving the experiments section and highlighting the significance of our findings to be helpful.
>
> ## Originality
>
> Please also see our comments on paper contributions and insights in the general response.
>
> ### Q2: Effect of the choice of pre-trained backbone for the multi-view encoder?  How much worse would a DINO or randomly initialized model do?
>
> In the revision, we have included additional experiments for different initializations of the shape encoder in Appendix A.2.3. We found that using a different initialization (OpenAI checkpoint) for the shape encoder and frozen CLIP model (OpenCLIP checkpoint) results in a drop-off in performance. This suggests that using the same initialization helps in making the fine-tuning smoother as the output latent space is similar as well as preventing overfitting on the frozen target text and image embeddings. However, we believe that this is a constraint of the 3D dataset size. And that with larger datasets in the future, our model can benefit from even more powerful 2D foundational models like DINO.
>
> ## Quality
>
> During failure case analysis, we observed incorrect labels in Objaverse-LVIS, primarily due to overly fine-grained categories within broader classes. For example, objects labeled as “pencil,” “pen,” or “stylus” were often misclassified within this group, and distinctions between objects in “bible” and “book” were visually indistinguishable. These errors mainly arise within hierarchical categories from what we observed, making Top3 and Top5 accuracy more indicative of the model's performance, where we also surpass previous work. Overall, we believe that reporting performance across a diverse range of datasets (Objaverse-LVIS, MVPNet, Text2Shape, ScanObjectNN, and ScanNet) highlights the significance of our model's capabilities.
>
> ## Clarity
>
> We have made changes to the writing in this section (L301-314) to make sure the flow of words is smoother.
>
> ### Q1:  Difference between 5F and 12F
>
> Yes, F here means the number of frames/multi-views used for evaluation. We have added a sentence to Table 2’s caption in the revision to clarify this.
>
> ## Significance
>
> ### Q3: Specific applications in robotics for which multi-view images are available in real time and where improved recognition from multi-view images would have a positive and significant impact?
>
> Please refer to the added citations in both the general response and the revised related works for potential robotics applications.
> We believe our model can enhance tasks like visual language navigation [1, 2, 3], where CLIP models are already used for object embeddings, by leveraging multi-view representations to improve 3D understanding. These tasks often require real-time navigation with only sparse RGB views, a limitation our model addresses by accommodating single or multiple views as more frames of the environment are observed.
>
> [1] Wang  et al. "Find what you want: learning demand-conditioned object attribute space for demand-driven navigation." NeurIPS  2023.
>
> [2] Gadre et al. "Cows on pasture: Baselines and benchmarks for language-driven zero-shot object navigation." CVPR 2023.
>
> [3] Garg, Sourav, et al. "Robohop: Segment-based topological map representation for open-world visual navigation." ICRA 2024.

---

> > ### Comment · Reviewer_p9H4 · 2024-11-28
> > **Thank you for the response**
> >
> > Thank you for the response that adequately addresses my concerns. I will keep my original score.

---

> > > ### Author Response · Authors · 2024-11-29
> > >
> > > Thank you for your time and insightful feedback. Your suggestions has helped us improve the paper, and we're glad our response has helped address your concerns.

---

### Official Review · Reviewer_ZsVt · 2024-11-04

**Soundness:** 4
**Presentation:** 3
**Contribution:** 2
**Rating:** 5
**Confidence:** 5

**Summary:**

The paper proposes a multi-view 3D understanding model based on contrastive learning. The model is a multiview image encoder encoding variable number of input frames and output a feature vector. The model is trained by constrasting against the corresponding clip and language embeddings.

**Strengths:**

- The paper is well-written. The figures and well-made and easy to understand. The overall presentation is good.
- The idea is simple and easy to understand. The architecture proposed might be reused for other multiview image tasks.
- The quantitative results are pretty strong and convincing.

**Weaknesses:**

- It's unclear to me why this task is important. The proposed method seems pretty incremental and I don't see a clear insights or surprising findings reviewed by the paper

**Questions:**

- Why is this an important task? What are some example applications of the proposed model? The paper seems to indicate that there are some applications in robotics. I'd like to see some examples of how the proposed technique can be used in robotic tasks and how exactly can current robotics methods can benefit from it.
- Can the model generalize outside of the training datasets? I didn't see any in-the-wild evaluation in the paper, which I feel is necessary as a metric for evaluation.
- Isn't it a little cyclic to use features from a frozen CLIP to train a multiview image encoder? Would the encoder benefit from some inter-view and inter-object contrastive losses? Wouldn't that allow the model to discover novel semantic and geometric information which might not have been captured by CLIP and text encoder?
- Can the learned multiview encoder be used for 3D reconstruction?

---

> ### Author Response · Authors · 2024-11-21
> **Response to Reviewer ZsVt**
>
> We thank the reviewer for taking the time to review our paper and provide valuable feedback. We are pleased that you found the paper well-written with clear figures, strong quantitative results, and promising potential for our architecture in other multiview image tasks.
>
> ## Insights and findings
>
> Our multi-view encoder provides an embedding of a 3D shape that is aligned to semantics (via text) and images, and our experiments show it outperforms prior work on zero-shot image-to-shape retrieval, text-to-shape retrieval and that it can be used for classification. The main finding of our work is that by using multi-view images, we obtain a representation that performs better than point clouds for these tasks (as also noted by R-p9H4).
>
> For more information, please see our comments in the general response.
>
> ## Q1: Importance of task and example applications
>
> In our paper, we focused on retrieval and classification experiments.  As 3D acquisition and content generation become better, we will have more and more 3D shapes, and shape retrieval itself is an important downstream application (just as image retrieval is).  Below we summarize various downstream applications that a shared representation across text, image, 3D allows for:
>
> - Robotics: Please see our general response for downstream robotics applications for our method.
> - Asset Retrieval: 3D scene generation already leverages CLIP models to retrieve 3D assets by using text queries and rendered images of 3D assets, which are then placed into the scene [1, 2]. By replacing the existing 3D asset embeddings in these databases with more powerful 3D representations like ours, we can achieve more accurate and contextually relevant object retrieval, ensuring better alignment with the input queries.
> - 3D LLMs: LLMs enable personalized, dynamic user interactions. Recent multimodal LLMs [3, 4] extend this to 3D data, using point cloud representations learned through similar frameworks. Integrating our model could improve performance and broaden input types, allowing queries with single or multi-view images (e.g., video frames), making real-world object interaction more accessible than using point clouds.
>
> ## Q2: Generalization outside of training set
>
> In our experiments, we included evaluations on two datasets that were not used in training: ScanObjectNN in Table 2 and ScanNet in Table 13 which we also outperform prior point cloud works. These evaluations are also in the wild as these are from scans or RGB frames of real-world scenes.
>
> In our revised manuscript, we also provide qualitative examples of image-to-shape retrievals from “in the wild” images (Figures 9 and 10 in Appendix A.4.1)
>
> ## Q3: Benefit of inter-view and inter-object contrastive losses
>
> We previously experimented with adding separate inter-view and inter-object contrastive losses but observed a significant drop in performance. We hypothesize that this is because inter-view images, processed by our MVA mechanism, share information in their embeddings. As a result, positive pairs become trivially easy to identify since the embeddings inherently reflect that they originate from the same object. Overall, we found the frozen CLIP embeddings more robust to learn from, while other contrastive losses often lead to overfitting.
>
> ## Q4: Using the learned multiview encoder be used for 3D reconstruction
>
> Yes, existing models, such as those based on foundation models like DINO or DINOv2, have been used for 3D reconstruction using single view [5] or multiple views [6].
> Our method offers distinct advantages over approaches like DINO or the original CLIP. Specifically, our method supports an arbitrary camera pose and number of views embedding with inter-view contextual understanding. For future work, additional losses could be explored to enhance structural features, similar to the strategies employed in DINO.
>
> [1] Huang et al. “Aladdin: Zero-Shot Hallucination of Stylized 3D Assets from Abstract
> Scene Descriptions.” arXiv preprint arXiv:2306.06212 (2023).
>
> [2] Fu et al. “AnyHome: Open-vocabulary generation of structured and textured 3D homes.” ECCV 2024.
>
> [3] Qi et al. "ShapeLLM: Universal 3D object understanding for embodied interaction." ECCV 2024.
>
> [4] Tang et al. “MiniGPT-3D: Efficiently aligning 3D point clouds with large language models using 2D priors.” ACM International Conference on Multimedia. 2024.
>
> [5] Hong et al. “LRM: Large Reconstruction Model for Single Image to 3D.” ICLR 2024.
>
> [6] Liu et al. “One-2-3-45++: Fast Single Image to 3D Objects with Consistent Multi-View
> Generation and 3D Diffusion” CVPR 2024

---

### Official Review · Reviewer_sfYJ · 2024-11-04

**Soundness:** 2
**Presentation:** 3
**Contribution:** 2
**Rating:** 6
**Confidence:** 4

**Summary:**

This paper introduces Duoduo CLIP, a model for 3D representation learning that captures object shapes from multi-view images instead of point clouds. By leveraging multi-view images, the model uses 2D priors from pre-trained CLIP models to fine-tune on 3D data. The proposed cross-view attention module aggregates information across different views, and the model is both permutation-invariant and pose-free. Compared to point cloud-based methods, this multi-view approach significantly reduces GPU training requirements and demonstrates improved performance on fine-grained text-to-shape retrieval tasks.

**Strengths:**

1. The paper is clearly written and easy to follow. Extensive experiments are conducted for point cloud classification and text-based retrieval.

2. The proposed method achieves state-of-the-art performance on these tasks while significantly reducing computational costs.

**Weaknesses:**

1. The zero-shot CLIP baseline is insufficient to demonstrate the proposed model’s advantages, as the test set is specific to 3D renderings, while the original CLIP model is primarily trained on natural images. To provide a fair comparison, the authors should fine-tune the original CLIP model using the same datasets and training strategies as the proposed model, and report the results.

2. Representing 3D shapes with multi-view images is one of the most straightforward approaches and has been used in previous works, such as ULIP2, to align the feature spaces of text, image, and point cloud representations. The purpose of such works is not solely to achieve high classification performance, but to deepen understanding of raw 3D representation beyond using just images. The original CLIP model, even without fine-tuning or special design, already serves as a strong baseline. This paper focuses on multi-view representation and achieves moderate improvements in basic classification and retrieval tasks by training on large-scale images, but this is quiet straight forward and it does not introduce new ideas or insights to the field. The work would be more compelling if the authors explored capabilities *beyond* the scope of those point cloud representations, such as scene-level multi-view data, to better demonstrate the necessity and potential upper bounds of the proposed approach.

3. Have the authors investigated the effect of different view sampling strategies? Unlike point cloud representations, which capture the entire shape, multi-view representations may be influenced by both the number and sampling strategy of views. While the impact of view count is examined in the paper, it would be beneficial if the authors also explored the impact of different sampling strategies, such as front versus back views, or uniformly sampled versus randomly sampled views.

4. Environmental lighting can significantly affect the appearance of 3D renderings by introducing shading, reflections, and slight color variations. It would be useful to conduct experiments examining how these lighting conditions impact the model’s performance.

5. Maybe add some more downstream tasks like image based retrieval and the concept combination as in Figure 1 of OpenShape.

**Questions:**

1. In Figure 6, the second row describes "a brown chair with a round back and thin metal base," but the ground truth appears to show a dark green or gray chair. It’s unclear whether this discrepancy is due to color shifts introduced during the rendering process or if it reflects a lack of quality control in the dataset.

---

> ### Author Response · Authors · 2024-11-21
> **Response to Reviewer sfYJ**
>
> We appreciate that the reviewer took the time to read our paper carefully, and found our work to be “clearly written and easy to follow”, with “extensive experiments” and state-of-the-art performance at reduced computational costs.
>
> ## W1: Zero-shot CLIP baseline insufficient
>
> We have revised the paper to include the fine-tuning CLIP baseline (FT B-32) as suggested, to Table 2, Table 3, and Table 4, with training details clarified in Section 4.2.1 (L318-320). FT B-32 mimics CLIP's single-image training while keeping other settings unchanged. Fine-tuning improves its LVIS top-one accuracy from 35.7 (zero-shot) to 50.1/53.0, but our model still outperforms it by 2.2–2.7 points. On ScanObjectNN and MVPNet, our model leads by 11.2 points, and while results on Text2Shape are close, our default model performs better. This shows that our multi-view training and MVA layers are effective in leveraging additional views to enhance performance across diverse datasets, outperforming the FT B-32 baseline in all cases.
>
> ## W2: Impact of using multi-view images for 3D
>
> We thank the reviewer for the valuable feedback. We refer the reviewer to the general response on paper contributions and insights for a detailed discussion.
>
> While our primary focus is on exploring the underutilized multi-view representations for 3D understanding and investigating efficient tuning strategies that preserve generalization, we agree that scene-level multi-view exploration is crucial for applications such as robotics. Although this lies beyond the scope of our current work, we see it as a natural and important direction for future research.
>
> ## W3: Effect of different view sampling strategies
>
> In Appendix A.1.2, we explore different camera sampling strategies for LVIS objects. Using Zero123's script, which samples camera poses uniformly across a sphere, yielded the best results, as diverse angles provide more comprehensive object information. Constraining views to the upper or extended hemisphere performed worse. We note Objaverse's lack of consistent front views across objects limits experimentation on sampling specific directions (e.g. front, back).
>
> ## W4: Effect of environmental lighting
>
> Our rendering script simulates diverse lighting by placing area lights at random locations with varying intensities, helping the model adapt to various scenarios. Rendering with environment maps for LVIS reduced performance likely due to the gap between the rendering used in training. Re-rendering the entire Objaverse dataset under diverse lighting conditions is computationally prohibitive for us due to its size and substantial storage requirements. Instead, we used Zero123's renders, which provided sufficient lighting and camera variation for our experiments. Developing efficient methods to bridge the gap between synthetic and real-world lighting remains critical but is beyond the scope of this study.
>
> ## W5: Additional downstream tasks
>
> We have added additional results for downstream tasks in the revision, including image-based retrieval (Figures 9 and 10 in Appendix A.4.1) and concept mixing (Figure 11 in Appendix A.4.2), as suggested by the reviewer.
>
> **Image-based retrieval.** For this task, we tested two indoor scene images: one sourced from the web and another generated by a text-to-image diffusion model, both serving as out-of-distribution data. For each cropped region, we retrieved the top 3 matching objects from the Objaverse LVIS dataset (46k objects). As shown in Figures 9 and 10, Duoduo CLIP consistently retrieves objects with strong semantic relationships, demonstrating its ability to understand high-level concepts despite the absence of visually identical matches.
>
> **Concept mixing.** We followed OpenShape’s approach of identifying the object in Objaverse that maximizes similarity to two objects being combined. As shown in Figure 11, our model effectively blends geometric concepts (e.g., pumpkin + carriage = pumpkin carriage) and semantic concepts (e.g., bird + bus = plane), showcasing its understanding of functional attributes, such as the wings of a bird enabling flight and the transportation role of a bus.
>
> ## Q1: Rendering of chair in Figure 6
>
> Thank you to the reviewer for pointing out the discrepancy. Upon closer inspection, we identified that this issue stems from a texture problem specific to this object. Re-examining the mesh revealed that the object displays different textures when loaded in different software, such as Blender or MeshLab.

---

> > ### Comment · Reviewer_sfYJ · 2024-11-29
> > **Thank you for the response**
> >
> > Thanks to the authors for their thorough rebuttal, which addresses most of my concerns. I encourage the authors to include the results from the fine-tuned CLIP baseline in the table, as this would provide a fair comparison with the proposed architecture. I will raise my rating.

---

> > > ### Author Response · Authors · 2024-11-29
> > >
> > > Thank you for your thoughtful feedback. We’re glad our rebuttal addressed most of your concerns. Following your suggestion, we have included the results from the fine-tuned CLIP baseline in Tables 2, 3, and 4 to ensure a fair comparison with the proposed architecture. If the paper is accepted, we will also incorporate these updates into the final version.

---

### Author Response · Authors · 2024-11-21
**General Response**

We thank the reviewers for taking the time to read our paper and provide valuable feedback.
All reviewers recognized that our paper was well-written and clear, with thorough experiments, and that our method “achieves state-of-the-art performance” (sfYJ, fQzu) while “significantly reducing computational costs” (sfYJ,fQzu), and that our results are “strong and convincing” (ZsVt).  Reviewers also noted that our method is “simple and easy to understand” (ZsVt), can “take advantage of large-scale pretrained representations” (p9H4) and the architecture could be reused (ZsVt).  Here we summarize the revisions we made to the paper, and include general responses addressing the contribution of our work and potential applications.  We provide more detailed responses to specific questions under each reviewer thread.

## Revisions

Based on feedback from the reviewers, we have added to the appendix examples of additional downstream application - image-to-shape retrieval on “in-the-wild” images of scenes, examples of context mixing, and experiments showing the effect of different initializations on the shape encoder.  We also added fine-tuning of CLIP as another baseline to our experiments, provided  clarifying details, including a discussion and comparison of the related work MixCon3D, as well as minor text edits.  Main edits are marked in magenta.

### Contributions

- Our straightforward and efficient method outperforms previous approaches that focused on scaling up models, while also addressing the underexplored potential of extending 2D priors with multi-view training for text and 3D understanding.
- We make improvements over SOTA methods that are predominantly point cloud based. We hope our method will reinvigorate interest in multi-view images as a viable option, offering better reuse of 2D priors.
- Our method supports applications with sparse views (more examples below), unlike point cloud methods requiring complete scans, and flexibly handles any number of input views, making it adaptable to various tasks.

Our work not only outperforms previous methods and demonstrates better zero-shot generalization but also leverages the underexplored multi-view representation for text and 3D understanding, addressing the gap in research while enabling diverse applications.

### Paper Insights

- We conduct extensive experiments on various factors affecting efficiency such as fine-tuning layers, number of multi-views during training and the MVA mechanism. Allowing for further scaling to larger models in the future.
- We also examine how these factors impact the preservation of the original CLIP model's generalization capabilities.
- Our method demonstrates better zero-shot generalization on unseen datasets compared to point cloud-based methods, highlighting the effectiveness of reusing 2D priors and the need for even larger datasets in the future.
- We also train on the real-world MVImgNet dataset, whereas previous point cloud methods primarily focus on synthetic data.

Our experiments provide new insights for the 3D community, showing that DuoduoCLIP, using only multi-view images, outperforms point cloud-based methods (Table 2). To our knowledge, this is the first demonstration of such results on these tasks, prompting further exploration of when multi-view images suffice and when point clouds offer unique advantages. While point clouds excel in geometric understanding and images in fine-grained appearance, we emphasize the importance of choosing the right representation for the scenario. Our aim is to highlight the underexplored potential of multi-view representations in 3D understanding for suitable applications.

---

> ### Author Response · Authors · 2024-11-21
> **Robotics Applications**
>
> ## Potential robotics applications (ZsVt, p9H4)
>
> In robotics, aggregating information from multi-view images is crucial as robots often capture different object views through multiple cameras or movement. Many methods use CLIP to locate objects based on language, such as identifying a “yellow toy animal” or navigating to a “wooden table.” Typically, object detectors aggregate CLIP image features (e.g., by averaging) to match text queries. DuoduoCLIP offers an alternative by extracting features from multi-view images directly (either from multiple cameras or from views over time). Additionally, we provide specific examples and citations for robotics applications that can potentially benefit from our work listed here and added the citations in the revision:
>
> - CLIP models are widely used in visual language navigation (VLN) tasks, such as encoding observed objects or image segments [1, 2, 3]. These tasks could benefit from our model's ability to encode multiple observed frames simultaneously in real-time, providing improved understanding and contextual awareness of objects within a scene.
> - There is also potential for applications such as for encoding observations for textual robotic manipulation [4] and building semantic field building for robot navigation [5].
> - Another application for robotics (that does not require real-time processing) is retrieving assets to build simulation environments for training and exploring new task setups [6,7,8].  Again, it is common to use CLIP scores for retrieval, but our DuoduoCLIP can serve as a drop in replacement.
>
> [1] Wang  et al. "Find what you want: learning demand-conditioned object attribute space for demand-driven navigation." NeurIPS  2023.
>
> [2] Gadre et al. "Cows on pasture: Baselines and benchmarks for language-driven zero-shot object navigation." CVPR 2023.
>
> [3] Garg, Sourav, et al. "Robohop: Segment-based topological map representation for open-world visual navigation." ICRA 2024.
>
> [4] Shridhar et al. "CLIPort: What and where pathways for robotic manipulation." CoRL 2021.
>
> [5] Shafiullah et al. "CLIP-Fields: Weakly supervised semantic fields for robotic memory." RSS 2023.
>
> [6] Yang et al. “Holodeck: Language Guided Generation of 3D Embodied AI Environments.” CVPR 2024.
>
> [7] Wang et al. “RoboGen: Towards Unleashing Infinite Data for Automated Robot Learning via Generative Simulation.” ICML 2024.
>
> [8] Wang et al. “Architect: Generating Vivid and Interactive 3D Scenes with Hierarchical 2D Inpainting.” NeurIPS 2024.

---

### Meta-Review · Area_Chair_mJu6 · 2024-12-14

**Metareview:**

The authors introduce a method to obtain shape embeddings for shape retrieval from multi-view images of the object. It is demonstrated that the method clearly outperforms previous retrieval methods in retrieval quality and efficiency. The paper received positively leaning reviews, slightly above the accept score, with one reviewer questioning the purpose of the solved task.
After taking a closer look at the paper I decided to agree with the majority and recommend to accept the paper.

**Additional Comments On Reviewer Discussion:**

During the discussions, the authors addressed many concerns of the reviewer well, leading to one of them increasing to an accept score. The negatively leaning reviewer did not react further and did not participate in the discussion. I found his single concern about relevancy to be addressed well and I would consider the solved task relevant.

---

### Decision · Program_Chairs · 2025-01-22

Accept (Poster)